# A Comprehensive Review on Monkeypox Viral Disease with Potential Diagnostics and Therapeutic Options

**DOI:** 10.3390/biomedicines11071826

**Published:** 2023-06-26

**Authors:** Ali A. Rabaan, Seham A. Al-Shwaikh, Wadha A. Alfouzan, Ali M. Al-Bahar, Mohammed Garout, Muhammad A. Halwani, Hawra Albayat, Norah B. Almutairi, Mohammed Alsaeed, Jeehan H. Alestad, Maha A. Al-Mozaini, Tala M. Al Ashgar, Sultan Alotaibi, Abdulmonem A. Abuzaid, Yahya Aldawood, Abdulmonem A. Alsaleh, Hani M. Al-Afghani, Jaffar A. Altowaileb, Abeer N. Alshukairi, Kovy Arteaga-Livias, Kirnpal Kaur Banga Singh, Mohd Imran

**Affiliations:** 1Molecular Diagnostic Laboratory, Johns Hopkins Aramco Healthcare, Dhahran 31311, Saudi Arabia; 2College of Medicine, Alfaisal University, Riyadh 11533, Saudi Arabia; 3Department of Public Health and Nutrition, The University of Haripur, Haripur 22610, Pakistan; 4Department of Commitment Management, Directorate of Health Affairs in the Eastern Province, Dammam 31176, Saudi Arabia; 5Department of Microbiology, Faculty of Medicine, Kuwait University, Safat 13110, Kuwait; 6Microbiology Unit, Department of Laboratories, Farwania Hospital, Farwania 85000, Kuwait; 7Department of Laboratory, Dhahran Long Term Care Hospital, Dhahran 34257, Saudi Arabia; 8Department of Community Medicine and Health Care for Pilgrims, Faculty of Medicine, Umm Al-Qura University, Makkah 21955, Saudi Arabia; 9Department of Medical Microbiology, Faculty of Medicine, Al Baha University, Al Baha 4781, Saudi Arabia; 10Infectious Disease Department, King Saud Medical City, Riyadh 7790, Saudi Arabia; 11Infectious Disease Division, Department of Medicine, Prince Sultan Military Medical City, Riyadh 11159, Saudi Arabia; 12Immunology and Infectious Microbiology Department, University of Glasgow, Glasgow G1 1XQ, UK; 13Microbiology Department, Collage of Medicine, Jabriya 46300, Kuwait; 14Immunocompromsised Host Research Section, Department of Infection and Immunity, King Faisal, Specialist Hospital and Research Centre, Riyadh 11564, Saudi Arabia; 15College of Medicine, King Saud bin Abdulaziz University for Health Sciences, Riyadh 11481, Saudi Arabia; 16Molecular Microbiology Department, King Fahad Medical City, Riyadh 11525, Saudi Arabia; 17Medical Microbiology Department, Security Forces Hospital Programme, Dammam 32314, Saudi Arabia; 18Clinical Laboratory Science Department, Mohammed Al-Mana College for Medical Sciences, Dammam 34222, Saudi Arabia; 19Laboratory Department, Security Forces Hospital, Makkah 24269, Saudi Arabia; 20iGene Center for Research and Training, Jeddah 2022, Saudi Arabia; 21Microbiology Laboratory, Laboratory Department, Qatif Central Hospital, Qatif 32654, Saudi Arabia; 22Department of Medicine, King Faisal Specialist Hospital and Research Center, Jeddah 22233, Saudi Arabia; 23Escuela de Medicina-Filial Ica, Universidad Privada San Juan Bautista, Ica 11000, Peru; 24Escuela de Medicina, Universidad Nacional Hermilio Valdizán, Huanuco 10000, Peru; 25Department of Medical Microbiology and Parasitology, School of Medical Sciences, Universiti Sains Malaysia, Kubang Kerian 16150, Malaysia; 26Department of Pharmaceutical Chemistry, Faculty of Pharmacy, Northern Border University, Rafha 91911, Saudi Arabia

**Keywords:** monkeypox, Mpox, treatment, poxvirus, *orthopoxviruses*, infectious diseases, outbreak, endemic, viral diseases, zoonosis

## Abstract

The purpose of this review is to give an up-to-date, thorough, and timely overview of monkeypox (Mpox), a severe infectious viral disease. Furthermore, this review provides an up-to-date treatment option for Mpox. The monkeypox virus (MPXV) has remained the most virulent poxvirus for humans since the elimination of smallpox approximately 41 years ago, with distribution mainly in central and west Africa. Mpox in humans is a zoonotically transferred disease that results in symptoms like those of smallpox. It had spread throughout west and central Africa when it was first diagnosed in the Republic of Congo in 1970. Mpox has become a major threat to global health security, necessitating a quick response by virologists, veterinarians, public health professionals, doctors, and researchers to create high-efficiency diagnostic tests, vaccinations, antivirals, and other infection control techniques. The emergence of epidemics outside of Africa emphasizes the disease’s global significance. A better understanding of Mpox’s dynamic epidemiology may be attained by increased surveillance and identification of cases.

## 1. Introduction

Monkeypox (Mpox) is an uncommon disease caused by infection with the monkeypox virus (MPXV), with a mortality rate of 0–10% depending on the clade (geographically based), which was first identified in wild animals such as rats and primates before spreading to humans [1]. It was first detected in laboratory monkeys in 1958, and examining blood from African animals revealed Mpox infection in various African rodents [2]. The MPXV can infect a wide range of animals, such as a variety of squirrel species (rope squirrel and the Gambian pouched rat) [3,4]. There are still a lot of unclear questions about the MPXV’s natural history. Additional research is required to identify the precise reservoir(s) and clarify how the virus spreads among wild animals [5,6].

The first case of Mpox in humans was reported in 1970 in the Democratic Republic of Congo, where an outbreak of smallpox was already occurring. Before that, this virus outbreak was seen in monkeys in 1958, where pox-like disease was observed in the monkeys [7]. The MPXV is a double-stranded deoxyribonucleic acid (DNA) zoonotic virus belonging to the Poxviridea family’s orthopoxvirus genus [8,9]. The MPXV has been divided into two distinct clades, the west African (WA) clade and the central African or Congo Basin (CB) clade. The CB clade was formerly believed to be more infectious and to produce more serious illness. The geographic boundary between the two groups is Cameroon, which is the only country where both viral clades have been identified. The current outbreaks going on outside of Africa are due to WA clade, which is significantly less virulent than CB clade [10,11].

The rest of the world is also affected by Mpox, making it a severe public health concern because it is not limited to countries in west and central Africa [12]. In 2003, the United States (US) experienced its first Mpox outbreak, which was traced back to prairie dogs kept as pets that had been infected [13]. Rats and dormice were brought to the US from Ghana, where they shared a cage with the pets. Over 70 people in the US contracted Mpox due to this outbreak. There have also been reports of Mpox in Nigerians traveling to Israel in September 2018, the United Kingdom (UK) in September 2018, December 2019, May 2021, and May 2022, Singapore in May 2019, and the US in July and November 2021 [14]. Mpox outbreaks have been reported in various non-endemic countries since May of 2022. Currently, researchers are working to better understand the spread of infection, its origins, and the methods by which it spreads [15,16].

The release of movement control orders (MCOs) and the relaxation on travel restrictions after the coronavirus disease 2019 (COVID-19) pandemic could be one of the risk factors for a Mpox outbreak [17,18]. The other possibility could be the evolution in the virus’s genome, which might play a role in the emergence of Mpox [19]. In a prior investigation, modifications to the genome of the MPXV (MPXV-BY-IMB25241) collected in May, 2022 were described [20]. Additionally, the phylogenetic analysis of the recently emerged Mpox revealed that these viruses belong to the west African clade and formed a distinct cluster. The authors suggested further genomic characterization to compare the hypervariable regions of these emerged viruses to obtain more insights into their evolution, which may affect their emergence [21]. Most cases in the recent outbreak have been found among men who are gay, bisexual or men who have sex with men (MSM) [22]. The health authorities are asking MSM to be aware of the symptoms, particularly if they have recently had a new sexual partner [23].

The symptoms of Mpox often last between two and four weeks [24,25]. Life-threatening cases are more likely to affect children whose immune systems are less developed and whose medical conditions are more precarious [26,27]. In other cases, even worse consequences can be caused by underlying immunological weaknesses [28]. People younger than 40 or up to 50 years old are more vulnerable to Mpox today because of the global halt of smallpox vaccination campaigns following the disease’s elimination in most countries [29,30]. Secondary infections such as bronchopneumonia, sepsis, encephalitis, and corneal infection can all result from Mpox, which can cause blindness if not treated earlier [31].

After surveillance in non-endemic countries strengthens, the World Health Organization (WHO) expects to see an increase in the total number of Mpox cases. Actions that can be taken right now include educating individuals at risk of contracting Mpox and stopping the disease from spreading further. Most likely to contract Mpox are those who have had direct physical contact with an infected person when they are still sick [32]. As part of the WHO’s efforts to ensure the safety of healthcare workers and others, including cleaners, the WHO provides guidelines on how to protect frontline healthcare personnel [33,34]. The present review aimed to increase public awareness, offer technical support for the preventive measures, and influence readiness and reaction activities. The relevant literature was searched in the PubMed, Google Scholar, Science Direct, Web of Science and Scopus databases by using the keywords “monkeypox”, “monkeypox virus”, “Mpox” and “orthopoxviruses”.

## 2. Transmission

It is still unclear how Mpox spread to those early cases of the continents’ residents. The Centers for Disease Control and Prevention (CDC) warns doctors to be on the lookout for individuals with rash infections that could be Mpox, regardless of gender or sexual orientation [35]. There are many hypotheses that state that it is transmitted from animals to humans, but human-to-human transmission is also possible [36,37,38]. It can be spread by touching, aerosols, infected animal bite, or scratch [13,39]. In a recent research study, a rotating chamber measuring 10.7 L that was designed to fit within a Class-III biological safety cabinet was utilised to test the MPXV’s susceptibility to aerosolization. The airborne viruses were found using culture and PCR after up to 90 h. For PCR analysis and culture analysis, the viral concentrations decreased by one log and two logs, respectively, in the first 18 h. Between 18 and 90 h, the viral concentrations stayed the same, which shows that the MPXV may stay infectious in aerosols for longer than 90 h [40] (Figure 1).

An infected person’s respiratory droplets or skin sores or recently contaminated things might transmit the disease to another person through close contact [41]. For respiratory droplet particles to transmit, face-to-face contact is often necessary, which raises the risk for healthcare professionals, household members, and other close contacts of active patients. The number of person-to-person infections in the longest chain of documented transmission, however, has increased from 6 to 9 in recent years. Smallpox vaccinations have been discontinued, which may have contributed to a decrease in immunity in all areas [42]. Additionally, transmission can occur via placenta (which might result in congenital Mpox), as well as through intimate contact between mother and child after birth [43]. Mpox has a typical incubation period of 7 to 14 days. However, this can vary from 5 to 21 days [44,45,46] (Figure 2).

## 3. Pathophysiology of Monkeypox

There is a wide range of disease severity, from moderate to lethal. Diarrhoea and vomiting, ocular scarring and conjunctivitis, encephalitis, and bronchopneumonia have all been documented as a complication of Mpox [47]. A common long-term complication of bacterial superinfection is permanent pitting scarring. There have been reports of miscarriage and more severe illnesses among pregnant women [48]. For one to four days after the febrile prodrome with headache and tiredness, the patient will typically exhibit centrifugal development of deep well-circumscribed macularpapular, vesicular and pustular lesions. At each stage, lesions endure between one and three days. Before or during the rash, lymphadenopathy may develop [49].

The MPXV can majorly affect the host’s various organ systems, including the mucosal surfaces, skin, gastrointestinal tract, lymphatics, and lungs, which serve as protective barriers [50]. Airway inflammation and bronchopneumonia can cause skin exfoliation to be considerable, as well as the decreased willingness or ability to consume food or water [51,52]. Mpox can also lead to sepsis and other body organ dysfunction syndromes [53,54].

### 3.1. Bronchopneumonia 

Bronchopneumonia is a significant consequence of Mpox, as it was previously known as smallpox. However, it is not well understood [55]. A common outcome of the pulmonary assault of NHPs at different infectious doses is the development of localised necrosis of lung tissues, fulminant bronchopneumonia, and diffuse pulmonary consolidation. Several studies [56,57] found that intratracheal deposition of virus-containing aerosols significantly worsened respiratory distress and increased animal death. Although *Klebsiella pneumoniae* just recently emerged in one of the animals that succumbed to its illness, the animals did not often develop future bacterial infections [58,59].

The manifestations of Mpox in people are similar to those of smallpox, although milder [60]. A fever, headache, muscle aches, and tiredness are among the symptoms of Mpox. Mpox causes lymph nodes to swell (lymphadenopathy), whereas smallpox does not, and this is the fundamental distinction between the two [61]. The lesion of Mpox progress as macule formation, papule formation, vesicle formation, pustule formation, and scab formation [50,62].

Patients experiencing increased body temperature, headache, cough, rash, myalgia, and lymph node enlargement three weeks after contact with prairie dogs or Gambia giant rats should have their symptoms evaluated by a doctor to rule out the possibility of Mpox [63]. The state or local health offices should be notified at once if specific illnesses are detected in people or animals [64,65].

### 3.2. Clinical Manifestation Comparison

The symptoms of Mpox are quite similar to those that were observed in the past in individuals who were suffering from smallpox, despite the fact that Mpox is clinically less severe [39]. It is brought about by the MPXV, which is classified as an orthopoxvirus and is a member of the Poxviridae family of viruses. The Poxviridae family and Orthopoxvirus genus viruses consist of various viruses known to infect humans, including the MPXV, vaccinia virus (VACV), cowpox virus (CPXV), and variola virus (VARV) [66]. In west and central Africa, Mpox occurs mainly in older children, adolescents, and young adults. In two large epidemiology studies of Mpox outbreaks, the investigators observed a sizable number of coinfections of chickenpox (varicella) and Mpox [67]. For variola, the case fatality rate majorly varied depending on the manifestation of the disease; nonetheless, aggregate case-fatality rates ranging from 10 to 30% have been observed over a number of outbreaks. The severity of the condition was found to have a correlation with the amount of rash that was present and was also found to be adverse in youngsters and in women during pregnancy. Furthermore, Mpox can also closely resemble a genital herpes simplex infection. Indeed, a patient can have both infections at the same time [68]. The clinical pictures of smallpox and varicella share a number of similarities with Mpox, which are given as a comparison in Table 1.

### 3.3. Secondary Co-Infections

Due to the significant skin irritation, there is a high risk of subsequent bacterial skin infections, which have been shown to be prevalent in 19% of unvaccinated individuals with smallpox [69]. Before the formation of crusts, the skin of infected individuals usually remains bloated, rigid, and excruciatingly painful [70]. The emergence of a second febrile phase, which takes place when skin lesions become pustular, has been linked to a worsening of the patient’s health status [71,72].

A retrospective analysis of hospital records for a total of 40 human cases of Mpox in Nigeria found that most patients suffered from fever and vesiculopustular skin eruptions that resolved on their own and did not need any treatment. The patients who were also co-infected with HIV type 1 had more severe lesions, a longer sickness, and more vaginal ulcers and subsequent bacterial skin infections. According to the findings, HIV-negative individuals did not have all of these characteristics [73].

The patients who survived after an illness were most likely to have permanent pitted scarring. The mortality rate of unvaccinated individuals is as high as 11%, and children are especially susceptible to severe sickness [28,74]. In a previous study, unvaccinated patients were shown to have 74% more severe problems and sequelae than immunized individuals (39.5%). A secondary lung infection has been found in patients who have developed respiratory discomfort or bronchopneumonia, generally at the end stage of their illness. By the second week of the disease, vomiting or diarrhoea can lead to severe dehydration. One patient had encephalitis, and another had septicaemia. Eye infections can lead to corneal scarring and irreversible vision loss if left untreated [75] (Figure 3).

## 4. Outbreak History of Monkeypox Virus Disease

Since 1970, there has been a steady increase in the number of reports of isolated outbreaks [8]. There has been a total of 35 separate outbreaks recorded outside of the Democratic Republic of the Congo (DRC), 20 of which have occurred since 2010 [2]. In 1996 and 1997, the outbreak of Mpox occurred in the DRC and raised questions about whether smallpox samples should be kept for comparison with other viruses such as Mpox [76]. A total of 71 cases of Mpox, including six deaths, were reported in 13 villages of Zaire between February and August 1997 [77]. In August 1997, when the outbreak was at its peak, the number of cases of secondary infections also increased. As the outbreak progressed, local health officials reported 170 new cases between March and May 1997 (58 in March, 52 in April, and 60 in May) to the WHO [77]. However, another study noted that some of these might be chickenpox cases [78]. Most cases of Mpox have been found in communities living in rural areas, in small villages (fewer than 1000 people), or in humid evergreen tropical forests, near or within the human–animal interface (HAI) places [79].

Furthermore, as a result of indirect or low-level exposure to sunlight, people who live in or near forest regions may develop subclinical illnesses [80,81]. After smallpox was eradicated, the first known human case may have occurred due to the subsequent exposure to infection. The disease was extremely rare and only found in the rain forests of western and central Africa, where it was first discovered in the 1970s.

Before the outbreak in the Midwest states in 2003, no cases of Mpox were reported in the US. Later, 71 suspected Mpox cases were reported between 16th May and 20th June 2003, of which 47 were confirmed cases [82]. The score for the burden of skin rashes, mortality and morbidity, hospitalization rates, and the severity of illness (a global score including the degree of incapacity, nursing care requirement, and burden of skin rash) were utilized to describe these diseases in humans [83,84,85]. Mpox viruses are divided into two main phylogenetic groups (those reported in west and central Africa) [86,87]. The severity of the disease varies between the clades reported in the US outbreak in 2003 and the west African clade [88].

Between 1970 and 2005, there were fewer than ten reported cases of the west African clade in Liberia, Sierra Leone, Nigeria, and Côte d’Ivoire, while the US outbreak reported 47 cases [3]. Humans and non-human primates were affected less when infected with west African clade Mpox. The outbreak in the US resulted in a large number of hospitalized patients, but no fatalities were reported [89].

A human Mpox case was reported to the WHO on 16 November 2021, by the IHR National Focal Point of the United States of America (USA) [90]. The patient had recently returned from a trip to Nigeria. When the skin rashes appeared, the patient was in Lagos, Nigeria [80]. The patient left Lagos, Nigeria, on November 6th for Istanbul, Turkey, and arrived in Washington, D.C., the US, on 7 November. While in isolation in Maryland, it was confirmed that the patient had not had any previous vaccinations against smallpox. *Orthopoxvirus*-generic and non-variola *orthopoxvirus* real-time (RT)-PCR assays on skin lesions were positive on 13 November at the Maryland laboratory of the Laboratory Response Network (LRN). The PCR testing performed on the identical two lesion specimens on November 16th confirmed the diagnosis of Mpox. More specifically, it was the strain of Mpox from the west African lineage that has been re-emerging in Nigeria since 2017 [7]. Because the patient had remained in Lagos the entire time he stayed in Nigeria, the infection source was unknown in this case. This case was the second time human Mpox was found in a visitor to the US [31]. Non-endemic countries such as Israel, Singapore, and the UK and Northern Ireland have also reported six cases of human Mpox among travellers from Nigeria since 2018. There is a big chance that tourists from endemic areas were infected due to their frequent travel around the world [83].

Endemic outbreaks were recorded in the twelve months prior to May 2022, with the highest number of infected patients in Congo [91]. Between 13 May and 21 May 2022, 92 cases of Mpox in non-endemic countries were reported to the WHO [92]. These 92 cases of Mpox were confirmed through PCR testing. On 17 June 2022, a total of 2103 cases were reported to the WHO with a mortality rate of 0.09% [93]. The confirmed Mpox cases increased to 3413 on 27 June 2022 [94]. As of 17 August 2022, 39,434 cases had been reported, with a mortality rate of 0.03%. On 13 May 2022, the WHO started collecting information about the outbreak from all around the world, which was reported at the state level and by health institutions.

### Suspected Cases

Individuals of any age who present with a sudden onset of severe rashes with fever (38.5 °C), backpain, myalgia, headache, asthenia, and lymphadenopathy in a non-endemic country could be considered as suspected cases. The following common causes of an acute rash are not supported by the clinical picture: allergy (e.g., to plants), bacterial skin infections, primary or secondary syphilis, disseminated gonococcal infection, and any other locally relevant frequent causes of the rash (papular or vecicular) [15].

A suspected case must have an epidemiological connection (including direct face-to-face exposure by healthcare workers without protective eyewear or respiratory equipment), close physical contact with skin or skin lesions, or contact with contaminated objects such as bedding, clothing, or utensils in the 21 days prior to the onset of Mpox symptoms [14]. After the start of worldwide surveillance, the WHO expects to see an increase in the number of cases of Mpox. Immediate measures include providing correct information to people who are most at risk of Mpox infection, halting the spread of the disease, and safeguarding frontline workers [83].

After recovering from the acute illness, retrospective instances discovered by active search may no longer exhibit the clinical signs of Mpox, but they may still show scarring and other sequelae [80]. In addition to current instances, retroactive cases may provide valuable epidemiological data. Although retrospective instances cannot be scientifically verified, their blood may be obtained and tested for anti-*orthopoxvirus* antibodies to help with case categorization [14].

Samples obtained from individuals suspected of having Mpox or from animals suspected of harbouring the MPXV should be handled carefully by qualified personnel working in appropriately furnished labs. When transporting samples to the testing facilities and packaging them, strict adherence to national and international rules on the transfer of infectious substances is required [2]. To address national laboratory testing capacities, careful planning is necessary. In order to reduce risk to laboratory staff and, when required, safely conduct laboratory tests that are needed for clinical treatment, clinical labs should be informed before samples from patients with suspected or confirmed Mpox are submitted [88].

## 5. Diagnostic Strategies

Usually, clinicians prefer a clinical diagnosis based on signs and symptoms and comparison with other diseases of the same category, such as smallpox and chickenpox. Confirmatory diagnosis should be a laboratory diagnosis, and some of the diagnostic tools with their pros and cons are described below.

### 5.1. Polymerase Chain Reaction (PCR)

Identification based on the presence of genetic material or DNA in a patient’s lesion material can be used to diagnose an active case [76]. To conduct the test, a specimen must be stored in a dark, cool setting to preserve the viral DNA [95]. Contamination problems necessitate the use of extremely sensitive assays designed to target the MPXV specifically. Equipment and reagents needed for these tests are expensive. A professional facility with experienced technicians is required to conduct this test [49,96].

### 5.2. Cell Culture

A chorioallantoic membrane can be used for live viral culture where this virus produces “pocks”. Lesion specimens can be used for such cultures [97,98]. The test takes several days to complete. The patient’s body might possess bacteria, which would make efforts at cultivation difficult. More characterization is needed for viral identification. It has to be performed in a big-capacity laboratory by skilled experts [99].

### 5.3. Immunohistological Techniques

This technique may be used to identify the antigens in biopsy specimens based on antigen–antibody interaction [97]. The application of this technique may be utilised to exclude or locate other suspected organisms. It must be carried out in a large laboratory by competent healthcare professionals [17].

### 5.4. ELISA

ELISA is used to detect the antibodies against the MPXV in human blood, which is basically a point-of-care diagnostic tool that uses a patient’s lesion material to quickly diagnose an active case. The test may be performed at room temperature with minimal training. The MPXV is not detected by this assay. There must be endemic testing carried out. PCR is more sensitive [63]. Serological cross-reactivity amongst *orthopoxvirus*es means that antigen and antibody detection methods cannot be used to confirm Mpox. Due to the lack of resources, serology and antigen detection procedures are not suggested for diagnostic use [17].

## 6. Treatment and Management

Studies of Mpox sickness in animals have recently shed light on the disease’s biology, proposing specific aspects of supportive therapy that may be both realistic and successful at enhancing patient outcomes. In order to test the effectiveness of antivirals and vaccines, researchers were motivated by a desire to duplicate a severe smallpox-like sickness. Up to this point, these studies have revealed therapeutically important information about Mpox pulmonary infection and other elements of the severe illness’s clinical picture, such as signs of poor prognosis [50]. Multi-organ functions in the host can be adversely affected by Mpox, which compromises the protective barriers of the skin and mucosa, causing a strong lymphatic inflammatory response and obstruction in the airways. Exfoliation can be substantial in cases of a heavy rash burden, putting patients at risk of dehydration and protein loss. Breathing can be severely restricted or perhaps impossible for a patient with bronchopneumonia, which is a severe inflammation of the airways [89].

Significant clinical signs of illness can also be caused by co-infections (malaria, varicella, HIV), as well as comorbidities (malnutrition). For patients in low-resource settings, a treatment plan must anticipate the likelihood of each of these outcomes occurring. In an ideal world, this evaluation would be based on scientifically sound criteria gleaned from extensive clinical research. In this way, resources can be used most effectively for the patient’s rehabilitation and the spread of virus is minimized [58].

### 6.1. Therapeutic Measures

Symptom relief, complications management, and long-term sequela prevention are all goals of optimal clinical care in the treatment of Mpox [10,100]. Fluids and food should be provided to patients to provide proper nutrition. It is important to treat any secondary bacterial infections that may arise. On the basis of results from both animal and human studies, the European Medicines Agency (EMA) granted a license to tecovirimat, an antiviral drug originally designed for smallpox [101]. Tecovirimat inhibits the intracellular release of the virus and is preferable because of its oral administration. Some other vital drugs include the following:Cidofovir can be used to inhibit viral replication by inactivation of DNA polymerase which can be administered intravenously, which could be nephrotoxic.Brincidofovir (CMX-001) is an orthopoxvirus nucleotide analogue DNA polymerase inhibitor and a lipid conjugate of the nucleotide analogue cidofovir and is indicated for the treatment of human smallpox disease (Figure 4).

### 6.2. Vaccination

According to a series of studies, more than 85% of Mpox cases were prevented by vaccination against smallpox [7,73,83]. The smallpox vaccine is the oldest vaccination and has undergone three generations of medical technology (first, second, and third generation). In order to eradicate smallpox, first-generation vaccines were widely dispersed between 1950 and 1970. Second-generation vaccinations, which were utilized in some regions of the smallpox eradication effort, were developed in chorioallantoic membranes or cell cultures for improved purity. Prior to the eradication of smallpox, third-generation vaccinations, which were based on attenuated strains of vaccinia, were developed and used [101].

Two types of smallpox vaccinations (modified vaccinia virus Ankara and ACAM2000) can be used for Mpox. The modified vaccinia virus Ankara (MVA) is one of the third-generation vaccines which are based on attenuated vaccinia viruses and is a live, nonreplicating vaccine. The MVA is sold under the trade name Jynneos (also known as Imvamune or Imvanex) in the USA. These are significantly less virulent and have fewer adverse effects. The MVA, a severely attenuated strain of the vaccinia virus, is a potential vector platform for the creation of viral-vectored vaccines due to its qualities of efficient transgene expression and safety profile. Traditional smallpox vaccines based on VV have a history of causing severe side effects in both susceptible and resistant individuals. MVA is a replication-deficient strain of VV that has been shown to be safe in people and immunocompromised animals [102]. The European Commission has approved the advertising of vaccines for adults, even those with impaired immune systems [103].

ACAM2000 is a smallpox vaccine that is capable of replication. This type of vaccine is recommended for only healthy, immunocompetent, and non-pregnant individuals with a high risk of exposure. ACAM2000 needs an infectious dosage to be administered through a bifurcated, sterile needle that is inserted about 15 times into the epidermis of the deltoid area of the arm. This vaccination has greater adverse effects than the MVA vaccine. A papule will appear two to five days after immunisation and develop into a vesicle after a few more days. The vesicle achieves its maximum size in one to two weeks, after which a scab develops and sheds after two weeks. Patients who receive the ACAM2000 vaccine often have moderate fever and lymphadenopathy within the first two weeks after vaccination [102]. This vaccine is a single-dose vaccine and can be used for both human and animals [104]. The T-cells are needed to stop progressive vaccinia in macaques that were injected with ACAM200 [102]. A study conducted by Smith et al., (2005) showed that antibodies are the main form of protection against Mpox provided by the current nonattenuated smallpox vaccination [105].

### 6.3. Prevention

Mpox can be contracted by close physical or sexual contact with an infected person. Personal safety measures include staying away from those who have symptoms of the illness, using safer methods of sexual encounters, washing hands with soap and water or using an alcohol-based hand rub and observing proper respiratory habits [15].

Raising public awareness of risk factors and educating people about the precautions they may take to reduce their exposure to the virus is the main strategy for preventing Mpox. The feasibility and appropriateness of vaccination to prevent and control Mpox are now being studied in scientific investigations. A number of countries have or are implementing vaccination policies for people who may be at risk, including laboratory workers, fast response teams, and health workers [83].

Controlling an outbreak necessitates constant monitoring and quick detection of any new cases. Close contact with an infected person is the most common way to contract the MPXV during an outbreak of human infection. There is an increased risk of infection for healthcare workers and their family members. Healthcare workers caring for patients with MPXV infection or specimens from them should follow normal infection control procedures. The patient should be cared for by those who have been vaccinated against smallpox [106].

#### Public Responsibility

If a person notices a red rash with a fever or a feeling of being sick, they should contact a doctor and be tested for Mpox [83]. Mpox cases should separate themselves until the scabs fall off and refrain from any kind of close physical contact. Symptoms of Mpox can be alleviated with supportive care at this time. The caretakers of a Mpox patient should take all necessary precautions, including wearing a mask and disinfecting any surfaces and objects they have touched [88].

### 6.4. Clinical Management

Standard contact and droplet precautions should be used by medical professionals who are treating the samples of patients infected with Mpox, whether the condition is suspected or proven. All healthcare facilities, including outpatient clinics and hospitals, are required to take these precautions [28]. Hand hygiene, proper handling of contaminated medical equipment, laundry, and garbage, and cleaning and disinfection of environmental surfaces are all examples of standard measures [15].

For low-resource settings to offer the minimum standard of supportive care for Mpox patients, it is anticipated that some expenditure, including investment in laboratory diagnostics, will be required. There is insufficient data to explain whether institutional investments in supportive medical services and therapies affect mortality and morbidity sufficient to justify the costs [80]. There is a need for research that examines how therapy intensity affects patient outcomes.

Stockpiling treatment plans for eye conditions, obtaining vaccinations, or supplying personal protective equipment are all examples of ways to contribute to the resources required to maintain a minimum level of care. Alternatively, even if an Mpox patient only has mild to moderate symptoms, they are still a transmission risk until the skin lesions heal [59].

#### Healthcare Workers’ Response

Prompt isolation in a private room with efficient ventilation systems, a separate restroom, and supervision is suggested in patients with suspected or proven infection. If single rooms are not accessible, the use of cohorts (confirmed with confirmed, suspected with suspected) might be employed to ensure a minimum of one meter between patients when an infection is suspected [28]. A gown, gloves, medical mask, and eye protection—face shield or goggles—are all examples of acceptable personal protective equipment. For patients who are in close proximity to healthcare personnel or other patients, they should be instructed to wear a medical mask, if they are able to do so. As an additional precaution, lesions might be covered with a sheet, bandage, or gown in order to reduce the risk of infection. It is imperative that all personal protective equipment be removed from the patient’s isolation area before leaving the facility [14,59].

### 6.5. Outbreak Surveillance

The term epidemic is used to define a tendency for a disease to affect a large number of individuals within a population, community, or region at a particular time. In places where there are no reported cases of such lethal diseases, a single case can lead to an epidemic later. Similarly, a single Mpox case can pose a significant public health threat, so clinicians should quickly notify national or local public health authorities of any suspected cases, regardless of whether they are simultaneously investigating other possible diagnosis. Cases should be recorded as soon as possible, following the case definitions above or national case definitions. All suspected and confirmed cases are required to be reported to the WHO via the International Health Regulations (IHR) National Focal Points (NFPs) in accordance with the IHR [3].

In order to give the best clinical treatment, isolate affected people to stop the spread of the disease, and identify and manage contacts, it is crucial to monitor and look into Mpox cases and clusters. The most commonly identified routes of transmission are used to tailor effective control and prevention methods [59,73]. The monitoring of rash-like sickness should be stepped up, and skin samples should be collected for confirmation testing as soon as possible [17]. Additional public health investigations are being carried out in non-endemic countries where cases have been discovered, including extensive case detection and contact tracing, laboratory analysis, clinical treatment, and isolation under supportive care [73]. To identify the clade(s) of the MPXV responsible for this outbreak, scientists have used genomic sequencing when it was possible [107]. A meeting of vaccine specialists is being held by the WHO [88].

## 7. Conclusions and Perspectives

It is important to remember that *orthopoxviruses* continue to organically exploit new ecological and geographical niches, especially in this era of increased concern about bioterrorist occurrences. Understanding the evolution of zoonotic *orthopoxviruses*, particularly the MPXV, may help us better understand how the variola virus, a pathogen with severe human pathogenicity and efficient transmission and a highly specialized (human) host range, evolved. For decision making, it is important to understand how these unique MPXV clades affect human disease, as well as how these clades may influence future outbreak responses and diagnostic tests, as well as possible outbreak-related decisions on vaccines and therapies. Mpox outbreaks in many non-endemic countries in May 2022, with no known links to endemic areas, are unusual. Further studies are needed to identify the source of the infection and prevent it from spreading further. To protect the public’s health, it is critical to investigate all possible transmission channels as the cause of this outbreak is discovered.

## Figures and Tables

**Figure 1 biomedicines-11-01826-f001:**
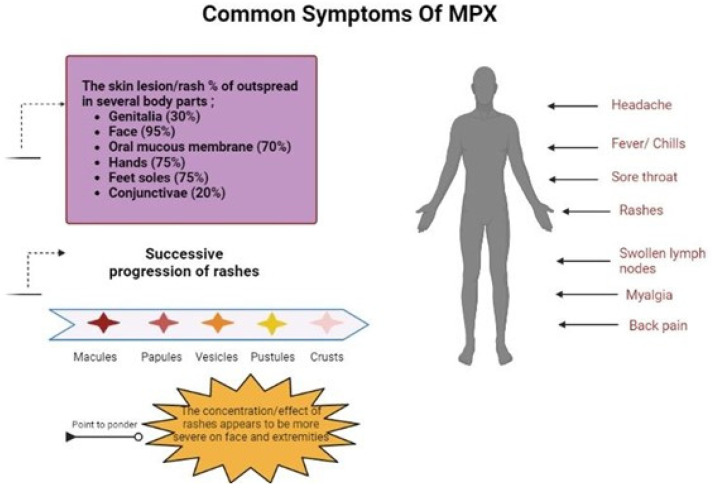
Common symptoms of Monkeypox virus diseases.

**Figure 2 biomedicines-11-01826-f002:**
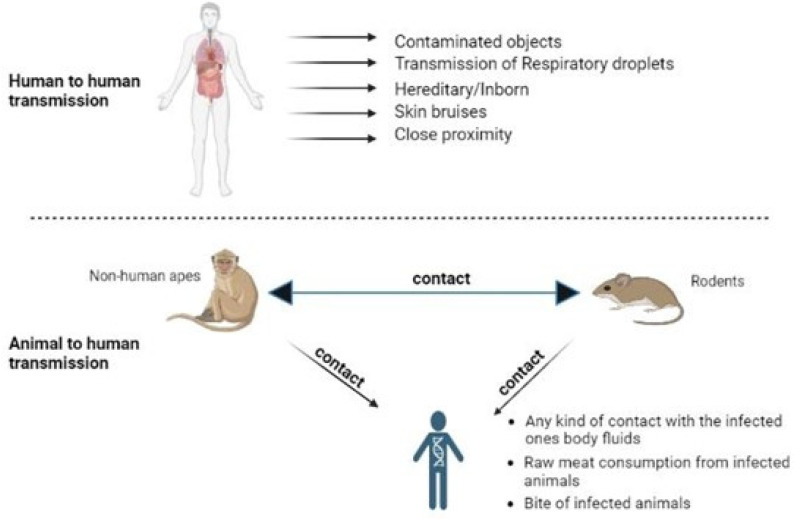
Transmission routes of monkeypox.

**Figure 3 biomedicines-11-01826-f003:**
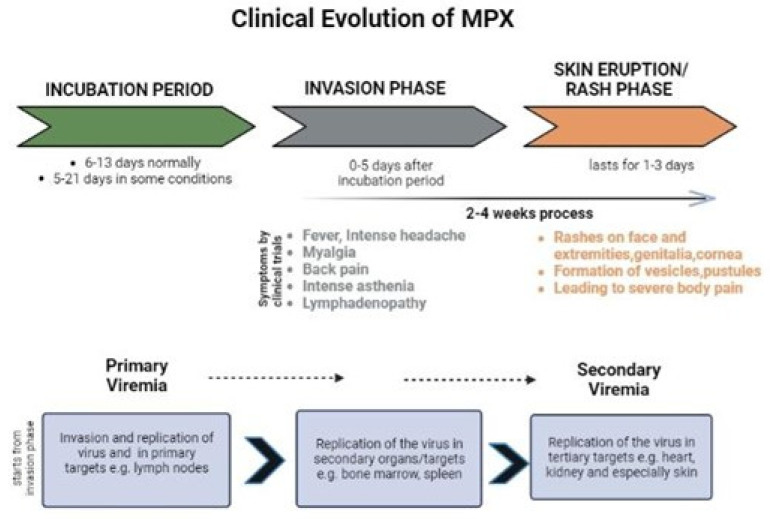
Clinical evolution of monkeypox.

**Figure 4 biomedicines-11-01826-f004:**
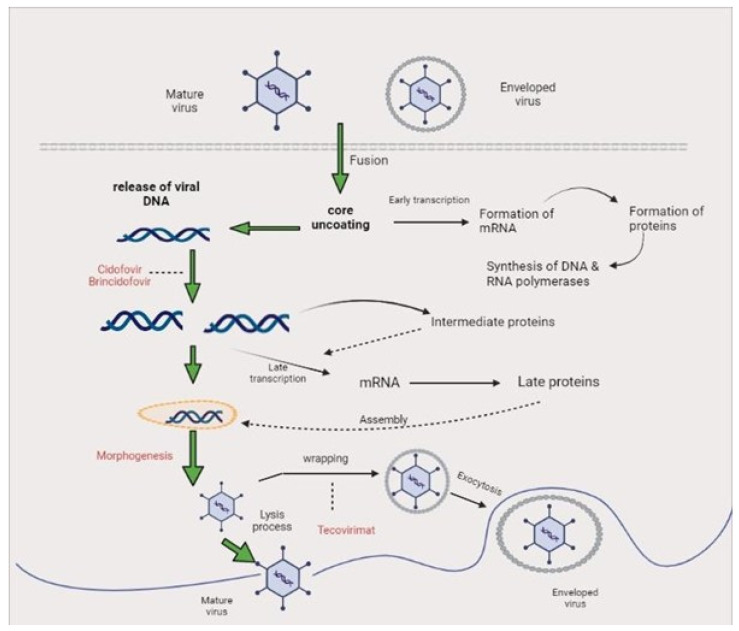
Monkeypox life cycle and mechanisms of action of antivirals.

**Table 1 biomedicines-11-01826-t001:** Clinical picture comparison.

Character	Smallpox	Monkeypox	Varicella
Viral cycle completion	28 days	28 days	21 days
Incubation time	2 weeks	2 weeks	3 weeks
Lesion inflammatory cycle	2–4 weeks	2–4 weeks	1–3 weeks
Body temperature	>40 °C	38.5–40.5 °C	38 ± 8 °C
Lymphadenopathy	Often	Often	Rare
Lesion	Centrifugal	Centrifugal	Centripetal

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
