# Peer review of "A Comprehensive Review on Monkeypox Viral Disease with Potential Diagnostics and Therapeutic Options"

_biomedicines, 2023, doi:10.3390/biomedicines11071826_

Round 1

Reviewer 1 Report

Thank you for sharing your review article on the Monkeypox viral disease.  Pleas find enclosed some comments that could help to improve the manuscript. Besides, your manuscript is very lengthy. Please consider shorten it for a potential reader to better follow the individual topics and understand your major findings. Please also include your data sources and the methodology, search criteria, and eligibility criteria applied to strengthen your work. 

L99-100: The content of the section "relaxation of travel restrictions following the coronavirus diseases-2019 (COVID) pandemic" isn't fully clear. Do you mean relaxation or intensification? 

L104-105: Why "the danger to the population of the UK is still considered minimal" in this content?

L132: To what time frame relates "the following days"?

L134: What do you mean by "for promptly indicated measures"; please clarify in your manuscript. 

L138: Pleas clarify in your manuscript what you mean by "it does not matter whether a patient has traveled or has other risk factors for monkeypox".

L143: Is "spread by touching, aerosols, infected animal bite or scratch" related to both, human-to-human and animal-to human transmission? Could it be transmitted also human-to-animal by the modes mentioned? 

L143-150: Does this paragraph related to the references 42 and 43? Please clarity in your manuscript. If it does not, please add the respective reference(s).

Figure 1, 2, 3 and 4: Are they taken from a publication or did you make them? Should those be original figures taken from another published manuscript, I believe you must state the respective reference(s). Please check the journal guidelines. Also, should you have taken the figures from a published manuscript and modified them  this must be stated here to in line with the journal's guidelines. 

L162-163: Doesn't your statement "the ability of monkeypox t spread, particularly via sexual transmission routes, is yet unknown" contradict your statement in L102-103?

L234: How is an outbreak defined in the context of Monkeypox?

L235-236: Seems you did not list all outbreaks in Table 2. As you wrote "some major outbreaks", based on which criteria did you decide to include/exclude them. 

L240: How was it confirmed that the deaths were due to monkeypox?

L243-244: To what does "only modest genetic variations were found among the 11 monkeypox specimens obtained between 1970 and 1979" relate? To the secondary infections? This needs to be clarified in your manuscript. 

L250-258: Consider presenting these age-related data in a different way. As data are presented now it is difficult to get the bigger picture/main message.

L266: Why suspected? Could the cases not be confirmed? 

L296: Here you start to structure this section. Could you think of a suitable title for the L233-295?

L301: Could the suspected cases be confirmed? If so, please state how.

L297-306: Are the reported cases equal to confirmed cases?

L307: Should the title be suspected case studies or suspected cases?

L318: Why only non-endemic countries?  

See above. 

Author Response

Reviewer 1

Comments and Suggestions for Authors

Thank you for sharing your review article on the Monkeypox viral disease.  Pleas find enclosed some comments that could help to improve the manuscript. Besides, your manuscript is very lengthy. Please consider shorten it for a potential reader to better follow the individual topics and understand your major findings. Please also include your data sources and the methodology, search criteria, and eligibility criteria applied to strengthen your work. 

Response: Dear reviewer, thank you for your valuable comments and suggestions. The manuscript has been thoroughly revised according to the comments from you and other reviewers. We must appreciate that, after addressing comments from you and other reviewers, the quality has been increased significantly. Furthermore, we have revised the manuscript for English proofreading and grammatical mistakes and more literature has been added to increase the scientific soundness of the manuscript. The search has been added at line 128-131.

L99-100: The content of the section "relaxation of travel restrictions following the coronavirus diseases-2019 (COVID) pandemic" isn't fully clear. Do you mean relaxation or intensification? 

Response: Line 99-101: The sentence has been revised.

L104-105: Why "the danger to the population of the UK is still considered minimal" in this content?

Response: The sentence has been removed starting from line 104-107 in the previous version of manuscript as at this time, the statement is no more correct.

L132: To what time frame relates "the following days"?

Response: The sentence has been removed starting from line 131-132 in the previous version of manuscript as at this time, the statement is no more correct.

L134: What do you mean by "for promptly indicated measures"; please clarify in your manuscript. 

Response: Line 128: The sentence has been corrected.

L138: Pleas clarify in your manuscript what you mean by "it does not matter whether a patient has traveled or has other risk factors for monkeypox".

Response: The statement has been removed from the Line 137 of previous version of manuscript as it was a repetitive sentence from line 99-101.

L143: Is "spread by touching, aerosols, infected animal bite or scratch" related to both, human-to-human and animal-to human transmission? Could it be transmitted also human-to-animal by the modes mentioned? 

Response: Transmission could potentially have occurred via fomite or through contact with infected thing originating from the rashes. Under the conditions of previous experimental and analytical findings, the airborne transmission route must be considered as a possible transmission mode. The previous findings and the analysis with aerosol dynamics showed that aerosols carrying MPXV could be present in environments where patients have resided and that airborne transmission of MPXV can occur.

L143-150: Does this paragraph related to the references 42 and 43? Please clarity in your manuscript. If it does not, please add the respective reference(s).

Response: Line 133-144: Correct reference has been cited.

Figure 1, 2, 3 and 4: Are they taken from a publication or did you make them? Should those be original figures taken from another published manuscript, I believe you must state the respective reference(s). Please check the journal guidelines. Also, should you have taken the figures from a published manuscript and modified them this must be stated here to in line with the journal's guidelines. 

Response: Dear reviewer, thank you for your observation about the originality of figures. All of the figures provided in the current review are original and made by the authors themselves. Hence, there is no need for a copyright permission from any of the paper/journal/publisher.

L162-163: Doesn't your statement "the ability of monkeypox t spread, particularly via sexual transmission routes, is yet unknown" contradict your statement in L102-103?

Response: Dear reviewer, we appreciate that you have noticed the contradiction between the statements. The statement at line 161-163 of the previous version of manuscript has been removed as it was not correct anymore. The sexual contact is already known for possible MXPV transmission.

L234: How is an outbreak defined in the context of Monkeypox?

Response: The outbreaks have been defined based on the data from CDC and WHO.

L235-236: Seems you did not list all outbreaks in Table 2. As you wrote "some major outbreaks", based on which criteria did you decide to include/exclude them. 

Response: Information in table 2 were taken from one of the previously published systematic review and meta-analysis entitled “A systematic review of the epidemiology of human monkeypox outbreaks and implications for outbreak strategy”. The authors have included situation reports, analyses of surveillance data cohorts, outbreak investigation reports, case reports, descriptions of surveillance data, press reports, case-control studies, cross-sectional study, mixed case-control and cross-sectional study and the modelling studies.

L240: How was it confirmed that the deaths were due to monkeypox?

Response: The information at line 240-242 was taken from the previous literature and it represents tha original data from the previous studies in 1997.

L243-244: To what does "only modest genetic variations were found among the 11 monkeypox specimens obtained between 1970 and 1979" relate? To the secondary infections? This needs to be clarified in your manuscript. 

Response: Line 242-243 from the previous version of the manuscript has been removed as it was not fitting in the paragraph. The respective paragraph was not representing the data of mutations/polymorphisms.

L250-258: Consider presenting these age-related data in a different way. As data are presented now it is difficult to get the bigger picture/main message.

Response: Dear reviewer, in order to make the manuscript more understandable for the reader, we have removed age-wise data from line 249-258 of the previous version of manuscript.

L266: Why suspected? Could the cases not be confirmed? 

Response: In 2003, there was an outbreak in the US. From the total suspected cases 47 were confirmed. The sentence has been corrected at line 263 of the revised version of manuscript.

L296: Here you start to structure this section. Could you think of a suitable title for the L233-295?

Response: The section 4 has been relabelled as “Outbreak history of monkeypox virus disease”.

L301: Could the suspected cases be confirmed? If so, please state how.

Response: The statement at line 299-300 from the previous version of manuscript has been removed as this statement is incorrect there.

L297-306: Are the reported cases equal to confirmed cases?

Response: The cases mentioned in the respective paragraph were of those which were reported to WHO and CDC. The actual scenario could be different. There might be more cases but was unable to report because of the less facilities in diagnosis or other issues.

L307: Should the title be suspected case studies or suspected cases?

Response: Section 4.1. The title has been changed as “Suspected cases”.

L318: Why only non-endemic countries?  

Response: The sentence means that when the WHO has started the surveillance in other countries where no monkeypox cases were reported, it can lead to more increase number of total cases worldwide. Line 312: the sentence has been revised.

Comments on the Quality of English Language

See above. 

Response: The manuscript has been thoroughly revised for English proofreading and grammatical mistakes.

Reviewer 2 Report

The authors present a review of monkeypox. Obviously monkeypox is an infectious disease of great current interest. There continue to be outbreaks in 2023. The review is comprehensive but there are occasional deficiencies, especially in recent references.  See comments below. 

1.Title and abstract and introduction. The WHO has recommended that the disease be called Mpox rather than monkeypox. The authors should mention this point in the Introduction. 

2.Transmission. the incubation period is described later in manuscript on lines 196-7. Suggest that the sentences about incubation period be moved forward and placed after line 163. Also, there are two recent manuscripts about short incubation periods among Europeans, averaging 7-8 days. Please mention these papers and add these 2 references: (a) Ward T, Christie R, Paton RS, Cumming F, Overton CE. Transmission dynamics of monkeypox in the United Kingdom: contact tracing study. BMJ. (2022) 379:e073153. doi: 10.1136/bmj-2022-073153; (b) Miura F, van Ewijk CE, Backer JA, Xiridou M, Franz E, Op de Coul E, et al. Estimated incubation period for monkeypox cases confirmed in the Netherlands. Euro Surveill. (2022) 27. doi: 10.2807/1560-7917.

3.Clinical manifestations section at line 204. This section is too short. It is true that mpox and chickenpox can closely resemble one another. Indeed, some patients have both diseases at the same time. Please write a few sentences about this fact. Please add a new reference about mpox and chickenpox by O. Khallafallah et al, VIRUSES 14 (12), 2800, 2022. PMID:36560805.

4.Clinical manifestations section. Mpox can also closely resemble genital herpes simplex infection. Indeed, a patient can have both infections at the same time. Please discuss this fact and cite article by K. Prabaker et al, American J. Trop. Med. Hyg. 107:1258-60, 2022. Title:Case Report: Symptomatic Herpes Simplex Virus Type 2 and Monkeypox Coinfection in an Adult Male.

5.Diagnostic strategies, line 338. Over 90% of cases are being diagnosed by PCR, not cell culture or immunohistochemistry. Therefore, please move PCR paragraph to the front of section as heading 5.1. Also please add a new reference about mpox pcr by J. Kanji et al, J. Assoc. Med. Microbiol. Infect. Dis. Can.8(1): 85-9, 2023. PMID: 37008576.

6.Conclusions section. Suggest that the last paragraph (lines 563-572) be deleted since the sentences are generalities that do not pertain to mpox.

Defer to the Editor.

Author Response

Reviewer 2

Comments and Suggestions for Authors

The authors present a review of monkeypox. Obviously monkeypox is an infectious disease of great current interest. There continue to be outbreaks in 2023. The review is comprehensive but there are occasional deficiencies, especially in recent references.  See comments below. 

Response: Dear reviewer, thank you for your valuable comments and suggestions. The manuscript has been thoroughly revised according to the comments from you and other reviewers. We must appreciate that, after addressing comments from you and other reviewers, the quality has been increased significantly. Furthermore, we have revised the manuscript for English proofreading and grammatical mistakes and more literature has been added to increase the scientific soundness of the manuscript. Some of the previous literature has been removed from the revised version of manuscript.

1.Title and abstract and introduction. The WHO has recommended that the disease be called Mpox rather than monkeypox. The authors should mention this point in the Introduction. 

Response: The word has been corrected throughout the manuscript.

2.Transmission. the incubation period is described later in manuscript on lines 196-7. Suggest that the sentences about incubation period be moved forward and placed after line 163. Also, there are two recent manuscripts about short incubation periods among Europeans, averaging 7-8 days. Please mention these papers and add these 2 references: (a) Ward T, Christie R, Paton RS, Cumming F, Overton CE. Transmission dynamics of monkeypox in the United Kingdom: contact tracing study. BMJ. (2022) 379:e073153. doi: 10.1136/bmj-2022-073153; (b) Miura F, van Ewijk CE, Backer JA, Xiridou M, Franz E, Op de Coul E, et al. Estimated incubation period for monkeypox cases confirmed in the Netherlands. Euro Surveill. (2022) 27. doi: 10.2807/1560-7917.

Response: Dear reviewer, thank you for your valuable suggestion. We have revised the content and the suggested relevant references has been cited.

3.Clinical manifestations section at line 204. This section is too short. It is true that mpox and chickenpox can closely resemble one another. Indeed, some patients have both diseases at the same time. Please write a few sentences about this fact. Please add a new reference about mpox and chickenpox by O. Khallafallah et al, VIRUSES 14 (12), 2800, 2022. PMID:36560805.

4.Clinical manifestations section. Mpox can also closely resemble genital herpes simplex infection. Indeed, a patient can have both infections at the same time. Please discuss this fact and cite article by K. Prabaker et al, American J. Trop. Med. Hyg. 107:1258-60, 2022. Title: Case Report: Symptomatic Herpes Simplex Virus Type 2 and Monkeypox Coinfection in an Adult Male.

Response:   Line 197-211: New literature has been added and the relevant references has been cited.

5.Diagnostic strategies, line 338. Over 90% of cases are being diagnosed by PCR, not cell culture or immunohistochemistry. Therefore, please move PCR paragraph to the front of section as heading 5.1. Also please add a new reference about mpox pcr by J. Kanji et al, J. Assoc. Med. Microbiol. Infect. Dis. Can.8(1): 85-9, 2023. PMID: 37008576.

Response: Dear reviewer, thank you for your valuable suggestion. We have revised the content and moved the PCR section to 5.1 and the suggested relevant reference has been cited to strengthen the statements.

6.Conclusions section. Suggest that the last paragraph (lines 563-572) be deleted since the sentences are generalities that do not pertain to mpox. 

Response: Dear reviewer, thank you for your valuable suggestion. We have removed the second paragraph of conclusion section.

Comments on the Quality of English Language

Defer to the Editor.

Response: The manuscript has been thoroughly revised for English proofreading and grammatical mistakes.

Round 2

Reviewer 1 Report

Thank you for sharing the revised manuscript. In principle, all my comments and suggested edits were addressed. One more issue I want to highlight and ask the authors to address is to check the journal's guidelines on how to report correctly on tables/figures taken from other published work. I am referring here in particular to Table 2 that can hardly be spotted as someone's other original work.

See above. 

Author Response

Reviewer 1

Comments and Suggestions for Authors

Thank you for sharing the revised manuscript. In principle, all my comments and suggested edits were addressed. One more issue I want to highlight and ask the authors to address is to check the journal's guidelines on how to report correctly on tables/figures taken from other published work. I am referring here in particular to Table 2 that can hardly be spotted as someone's other original work.

Response: Dear reviewer, we really appreciate your kind efforts to improve the quality of our manuscript. We would like to thank you because after addressing the comments from you and other reviewers, the quality of our manuscript has been significantly improved. Comments from you and other reviewers helped us alot. Furthermore, we want to mention that all of the figures proposed in the current review were originally made by us and don’t need any copyright permissions or references.

The information in table 2 of the last version was taken from previously published record, and for this we have cited the relevant references. This kind of information doesn’t need any ethical considerations except the references, as we were just mentioning that these cases were reported previously in different countries.

For ethical considerations, we have removed table 2 from the revised version of the manuscript as all of this information has already been mentioned in section 4. All of the relevant references have been cited in the respective records.